# Impact of Novel Active Layer-by-Layer Edible Coating on the Qualitative and Biochemical Traits of Minimally Processed ‘Annurca Rossa del Sud’ Apple Fruit

**DOI:** 10.3390/ijms24098315

**Published:** 2023-05-05

**Authors:** Anna Magri, Pietro Rega, Giuseppe Capriolo, Milena Petriccione

**Affiliations:** 1Department of Environmental, Biological and Pharmaceutical Sciences and Technologies (DiSTABiF), University of Campania Luigi Vanvitelli, Via Vivaldi 43, 81100 Caserta, Italy; anna.magri@unicampania.it; 2Council for Agricultural Research and Economics (CREA), Research Center for Olive, Fruits and Citrus Crops, 81100 Caserta, Italy; pietro.rega@crea.gov.it (P.R.); giuseppe.capriolo@crea.gov.it (G.C.)

**Keywords:** sustainable coating, enzymatic browning, fresh-cut fruit, antioxidant system, AA-GSH cycle

## Abstract

The color changes brought on by the enzymatic interactions of phenolic compounds with released endogenous polyphenol oxidase and the penetration of oxygen into the tissue has a significant impact on the commercialization of fresh-cut fruit, such as apples. This process causes a loss of quality in fresh-cut apples, resulting in browning of the fruit surface. By acting as a semipermeable barrier to gases and water vapor and thus lowering respiration, enzymatic browning, and water loss, edible coatings can provide a chance to increase the shelf life of fresh-cut produce. In this study, the effect of edible coatings composed of carboxymethylcellulose (CMC, 1%), sodium alginate (SA, 1%), citric acid (CA, 1%), and oxalic acid (OA, 0.5%) on fresh-cut ‘Annurca Rossa del Sud’ apple was studied. Four formulations of edible coatings, A. SA+CMC, B. SA+CMC+CA, C. SA+CMC+OA, and D. SA+CMC+CA+OA, were tested. Fresh-cut apples were dipped into different solutions and then stored at 4 °C, and physicochemical and biochemical analyses were performed at 0, 4, 8, and 12 days of storage. Results demonstrated that all four combinations improved the shelf-life of fresh-cut apple by slowing down the qualitative postharvest decay, total soluble solid, and titratable acidity. The browning index was highest in the control samples (82%), followed by CMC+SA (53%), CMC+SA+CA (32%), CMC+SA+OA (22%), and finally CMC+SA+CA+OA (7%) after 12 days of cold storage. Furthermore, coating application increased the bioactive compound content and antioxidant enzyme activities. Furthermore, the synergistic activity of SA+CMC+CA+OA reduces enzymatic browning, prolonging the postharvest life of minimally processed ‘Annurca Rossa del Sud’ apples.

## 1. Introduction

Apples are a valuable source of micronutrients, including vitamins, zinc, and iron, as well as phytonutrients, such as anthocyanins, flavanols, and flavonols, which have antioxidant activity [1]. By scavenging and eradicating cellular free radicals, these components in apples assist in controlling immunological responses and defending against chronic illnesses [2]. A substitute for the provision of fresh-like, extremely nutrient-dense, practical, and healthy commodities is minimal processing. Ready-to-eat apples have recently gained popularity for their convenience, portability, and ease of access and they satisfy your need for a healthy snack [3]. The safety and attractiveness of fresh-cut fruit may be compromised by mechanical bruising that occurs during processing and handling, which also causes an increase in respiration rates and several biochemical responses that underpin microbiological decomposition and quality decline [4]. The principal problem affecting the quality of fresh-cut apples is generally acknowledged to be enzymatic browning [5]. The enzymatic conversion of endogenous phenols to quinones by polyphenol oxidase (PPO) in the presence of oxygen is commonly acknowledged as the cause of the browning of fresh-cut fruit and vegetables [6,7]. 

Edible coatings and films are widely used to prolong the quality shelf life of many horticultural products [8,9]. The edible films and coatings function as semipermeable membranes that limit the flow of gases and water vapor, which is useful in slowing down the rate of respiration and water loss from the fruit and can subsequently slow the rate of physiological postharvest deterioration [10]. A variety of structural ingredients, including proteins, lipids, and polysaccharides such as cellulose, chitosan, starch, and pectin, can be used in the coating formulation due to the diversification of edible coating manufacture [11,12]. The building elements of the edible coatings are classified into hydrocolloids (proteins and polysaccharides), lipids, and composites [13]. Layer-by-layer (LbL) electrostatic self-assembly coating is a promising technology adopted to prolong the shelf-life of whole and fresh-cut fruits [14]. 

Sodium alginate (SA; E401) is defined as a “Generally Recognized as Safe” (GRAS) ingredient used in the food industry as an emulsifier, stabilizer, thickener, and gelling agent; it has a low price and great advantages. Carboxymethyl cellulose (CMC; E466) is a water-soluble polysaccharide with suitable biodegradable and edible film-forming properties that can be generated in big quantities and at low cost from a variety of sources [15]. Several studies evaluated the efficacy of hydrocolloids composed of carboxymethylcellulose or sodium alginate for the formulation of mono- and bi-layer coatings, which can increase the postharvest life of citrus [16], pistachio kernels [17], mango [18], ‘Kinnow’ mandarin [19], pear [20], strawberries [21], and walnut kernel [22]. Despite the fact that some studies have reported the antioxidant properties of films or coatings obtained by SA-CMC, little is known about their effect on food [23,24]. An edible coating realized by SA-CMC with Brewer Yeast has been used to prevent a qualitative decay and significantly increase the shelf life of grapes [25].

Edible coatings combined with organic acids represent a valid tool to preserve the quality and extend the storability of fresh product [26]. Oxalic acid (OA) is a naturally occurring substance that has a variety of physiological effects on plant tissues. Several studies showed that oxalic acid was involved in controlling tissue enzyme browning, inducing systemic resistance, retarding fruit ripening, suppressing the lipid peroxidation, and controlling qualitative decay [27,28,29,30]. 

Citric acid plays an important role in physiological and metabolic respiratory pathways [31]; furthermore, its copper chelating action inhibits the activity of polyphenol oxidase [32]. It has been reported that citric acid can prevent browning and extend the shelf life of several fruit crops [31,33,34].

However, fresh-cut apples are highly perishable, and several studies have been conducted to improve the postharvest life of minimally processed apple fruits using the following methods: alginate and gellan-based edible coatings [35], photodynamic bacteria inactivation with quercetin/pectine edible coating [36], corn zein-based water-soluble coating formulation impregnated with nisin [37], whey protein isolate/high methoxyl pectin-based carvacrol emulsions [38], and probiotic film based on gellan gum, cranberry extract, and *Lactococcus lactis* [39]. 

Currently, no other studies have been conducted on the synergistic effect of carboxymethyl cellulose and sodium alginate combined with oxalic and citric acid on the preservation of ready-to-eat apple or other fruit crops. Our study aimed to evaluate the effect of different layer-by-layer edible coatings on the quality and biochemical characteristics of minimally processed ‘Annurca Rossa del Sud’ apples during 12 days of cold storage.

## 2. Results and Discussions

### 2.1. Physicochemical Traits

Fresh-cut fruits and vegetables respond quickly to wounding stress, producing large amounts of ethylene and a rising respiratory rate, with loss of firmness and rapid qualitative decay [40]. In Table 1, the physicochemical parameters of coated and uncoated apple slices at 0, 4, 8, and 12 days of cold storage are reported. Weight loss is a key factor that influences the quality and storage life of fruits [41]. The weight loss significantly increased during storage in all samples (*p* ≤ 0.05), but tested coatings helped to reduce the moisture transfer from the fresh-cut fruit to the surrounding environment. CMC and SA-based treatment, alone or in conjunction with CA and AO, on fresh-cut apple fruits showed a reduction in the weight loss compared to control samples, demonstrating a synergistic effect of different compounds in the tested coatings (Table 1). Both CMC+SA+AO- and CMC+SA+CA+AO-coated apple slices showed significantly (*p* ≤ 0.05) lower weight loss compared to other samples at the end of storage (Table 1). Several studies have demonstrated that edible coatings produce a barrier layer on the fruit cutting surface, reducing the moisture transfer responsible for shrinkage, water loss, and decay [42,43]. In particular, in fresh-cut apples, the weight loss can be reduced by applying a bilayer probiotic edible coating containing carboxymethyl cellulose, zein, and *Lactobacillus plantarum* 299v [44] and an active coating composed of aloe vera gel or alginate and calcium lactate with ferulic acid [45].

The important commercial indicators of fruit quality are sugar content and acidity, which are related to fruit ripening and influence the organoleptic quality [46]. TSS increased throughout storage in all samples with a significant difference among coated and control fruits. CMC+SA+CA+AO-coated apple showed the lowest values (11.9 ± 0.06) at day 12 (Table 1), demonstrating the effectiveness of this coating in slowing down the moisture loss and fruit ripening with the breakdown of starch to sugar. Manzoor et al. [41] reported that the TSS of fresh-cut kiwifruit coated with alginate, carboxymethylcellulose, ascorbic acid, and vanillin was low compared to the control sample. Other authors also suggested that the TSS of the coated apple samples was stable during the 7 days of storage [44,45].

During storage, the changes in apple acidity affect the sugar/acid ratio responsible for apple flavor. At the end of storage, the lowest significant TA value was observed in the control sample, while it was higher for CMC+SA+AO and CMC+SA+CA+AO (Table 1). 

The decrease in TA content observed in the control sample is due to a higher respiration rate that caused the oxidation of organic acids and their decreases over time [47].

A significant increase in the pH value was detected in all samples with a lower value in the control compared to coated samples throughout storage (Table 1). Other studies reported that no significant change in the pH value has been observed in apple slices coated with bilayer probiotic, with pectin combined to essential oil and with a combination of ferulic acid with aloe vera gel or alginate during cold storage [44,45,48].

### 2.2. Secondary Metabolite Content

Phenolic compounds are found in abundance in higher plants. In response to damage stress, plants rapidly produce more phenols, which triggers an injury defense response for wound healing and repair. Processing procedures, including peeling, coring, cutting, and slicing, cause serious mechanical damage to fresh-cut fruits and vegetables, triggering the production of phenolic compounds to be activated [49]. In our study, total phenols (POL) were enhanced by LbL coating application, with or without antibrowning agents (Figure 1a). However, the POL of CMC+SA+OA+CA samples was higher than that of all other samples, with values of 213.17 ± 10.97, 249.19 ± 0.81, and 272.96 ± 2.79 mg GAE 100 g^−1^ FW at 4, 8, and 12 days of cold storage (Figure 1a). Compared with the control sample, CMC+SA+OA apples revealed an increase of about 25, 82, and 120% at three times analyzed, respectively (Figure 1a). On the contrary, the CMC+SA and CMC+SA+CA samples showed significantly higher values than the control, with a stable trend up to 8 days of cold storage, reaching maximum values of 193.63 ± 5.02 and 181.65 ± 4.76 mg GAE 100 g^−1^ FW, respectively, at the end of storage (Figure 1a). 

Flavonoid (FLA) production increased in all treated samples, while it decreased in the control sample (Figure 1b). FLA, during the 12 days of storage at 4 °C, in the control sample decreased from 60.24 ± 1.99 to 47.87 ± 4.23 mg CA 100 g^−1^ FW. In the coated samples, the values increased from 71.32 ± 1.30 to 81.27 ± 2.40 mg CA 100 g^−1^ FW in the CMC+SA treatment, from 70.20 ± 1.67 to 80.38 ± 1.93 mg CA 100 g^−1^ FW in the CMC+SA+CA samples, and from 75.05 ± 2.66 to 87.52 ± 1.36 mg CA 100 g^−1^ FW in the CMC+SA+OA apples at the end of storage (Figure 1b). As with POL, FLA showed the highest value in the CMC+SA+OA+CA samples with values of 103.64 ± 2.79, 113.35 ± 3.69, and 126.27 ± 4.21 mg CA 100 g^−1^ FW at 4, 8, and 12 days of cold storage, respectively (Figure 1b). 

Anthocyanins are water-soluble pigments that give fruits, flowers, and leaves a variety of appealing colors. In our study, anthocyanin content (ANT) decreased in the control samples, with a minimum of 2.14 ± 0.57 at 12 days of cold storage (Figure 1c). The CMC+SA sample showed a 2-fold higher value (4.52 ± 0.47 mg C3G 100 g^−1^ FW) than the control sample at the end of cold storage, while in the CMC+SA+CA apple, ANT, after an increase recorded at 4 days cold storage (2.96 ± 0.04 mg C3G 100 g^−1^ FW), decreased to 2.74 ± 0.19 at the end of the storage period (Figure 1c). However, the ANT of the CMC+SA+OA and CMC+SA+OA+CA samples was higher than all that of the other samples, with values of 3.89 ± 0.47 and 5.17 ± 0.15 mg C3G 100 g^−1^ FW for the first timing, 5.80 ± 1.05 and 7.98 ± 0.14 mg C3G 100 g^−1^ FW for the second timing, and 7.24 ± 0.38 and 10.15 ± 0.02 C3G 100 g^−1^ FW for the last timing (Figure 1c).

**Figure 1 ijms-24-08315-f001:**
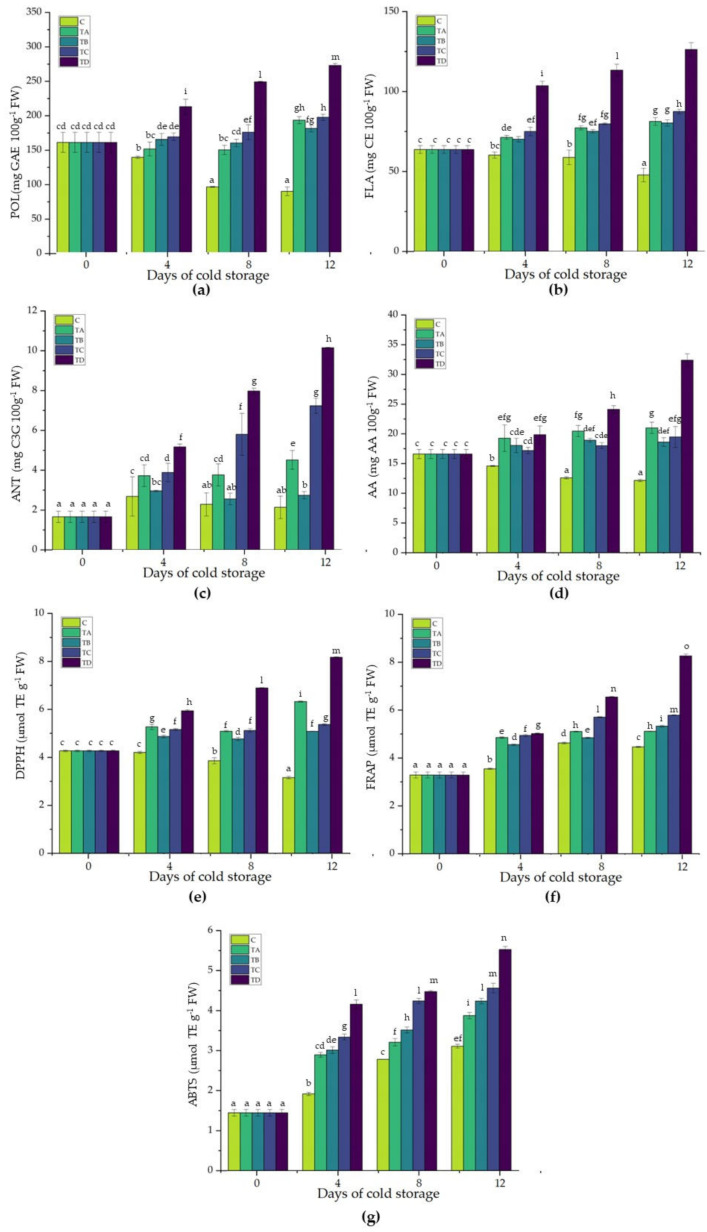
Total phenols (**a**), flavonoids (**b**), anthocyanins (**c**), ascorbic acid (**d**), DPPH (**e**), FRAP (**f**), and ABTS (**g**) in the control (C) and treated samples with CMC+SA (TA), CMC+SA+CA (TB), CMC+SA+AO (TC), and CMC+SA+CA+AO (TD) in fresh-cut ‘Annurca Rossa del Sud’ apple after 0, 4, 8, and 12 days of cold storage at 4 °C ± 0.5. Error bars indicate standard deviation. Means followed by the same letter do not differ significantly at *p* = 0.05 (Duncan’s test).

Fruit are well-known sources of bioactive compounds, with antioxidant properties conferring a lot of beneficial effects for human health [50]. The main phenolic substances found in fruit are phenolic acid, monophenol, flavonoids, and hydroxyl cinnamic acid derivatives. Additionally, flavonoids also comprise anthocyanins, flavonols, and isoflavones [40]. Several studies have demonstrated an increase in phenolic compound content in coated fresh-cut fruits due to the phenylalanine ammonia lyase activation, which is the first step in phenylpropanoid metabolism, after cutting [40,45,51]. Phenol synthesis is considered a defense mechanism activated against wounding and the levels of these compounds are the result of a balance between their biosynthesis and utilization rate [52,53]. An increase in phenolic substances in fresh-cut fruit can improve the functional value of these products [40].

Fruit growth, postharvest storage, and stress tolerance are all regulated by ascorbic acid, which also impacts fruit ripening [50]. In the present study, ascorbic acid content (AA) decreased in the control sample, with a minimum of 12.16 ± 0.21 mg AA 100 g^−1^ FW after 12 days of storage and a maximum of 14.59 ± 0.07 mg AA 100 g^−1^ FW after 4 days of storage (Figure 1d). In all treated samples, there is an increase in ascorbic acid. Samples coated with CMC+SA, CMC+SA+CA, and CMC+SA+OA showed an increase of 32, 24, and 18% after 4 days, 63, 50, and 43% after 8 days, and 73, 53, and 60% at the end of cold storage, respectively (Figure 1d). The sample coated with the CMC+SA+OA+CA complex coating showed an increase of 36, 92, and 166% in the three times considered compared to the control samples at the same times (Figure 1d). The ascorbic acid content reported herein (~17 mg/100 g DW at T0) is similar to values previously reported for the ‘Annurca Rossa del Sud’ apple [54]. The change in ascorbic acid content could be due to the activation of some genes involved in its recycling process following mechanical wounding [55]. In addition to its role as an antioxidant, ascorbic acid also has an important dietary value [56]. The increase in ascorbic acid content in coated fresh-cut apple could be due to the low oxygen permeability of the several coatings, which lowered the activity of the enzymes and prevented the oxidation of ascorbic acid combined with reduced water loss [57].

Total fruit antioxidant capability can be assessed using a range of techniques that can consider the various processes [57]. The total antioxidant capacity of coated and uncoated ‘Annurca Rossa del Sud’ apples was evaluated by 1,1-diphenyl-2-picryl-hydrazine (DPPH) radical, ABTS^•+^ radical cation, and Fe(III) reducing efficacy (FRAP). In general, the application of LBL coatings significantly improved the antioxidant activity of fresh-cut apples (Figure 1e–g). In particular, the lowest antioxidant activity using the DPPH method was recorded in the control samples, with values of 4.20 ± 0.04, 3.86 ± 0.13, and 3.16 ± 0.05 µmol TE g^−1^ FW at 4, 8, and 12 days of cold storage, respectively (Figure 1e). Compared to the control, the CMC+SA apples showed an increase in antioxidant activity of approximately 25, 32, and 100% at the three timings of storage (Figure 1e). A less pronounced, but still significant, percentage increase in the DPPH assay was also recorded in the CMC+SA+CA and CMC+SA+OA samples with values of about 16 and 23% after 4 days, 24 and 33% after 8 days, and 61 and 70% after 12 days of cold storage, respectively (Figure 1e). In the case of the CMC+SA+OA+CA samples, the DPPH values ranged from 5.94 ± 0.04 to 8.17 ± 0.03 µmol TE g^−1^ FW at the end of storage (Figure 1e). FRAP increased in all treated samples (Figure 1f). In the control samples, the FRAP value reached a minimum of 3.55 ± 0.02 µmol TE g^−1^ FW at 4 days and a maximum of 4.63 ± 0.04 µmol TE g^−1^ FW at 8 days of cold storage. FRAP showed a similar trend in the CMC+SA, CMC+SA+CA, and CMC+SA+OA samples, with statistically significant increases of approximately 37, 29, and 39% after 4 days cold storage, 10, 5 and 23% after 8 days, and 15, 19, and 30% at the end of cold storage compared to control samples (Figure 1f). The samples coated with the complex coating (CMC+SA+OA+CA) exhibited the highest FRAP value with a percentage increase of 41, 42, and 85% compared to the values found in the control samples at 4, 8, and 12 days of cold storage, respectively (Figure 1f). ABTS radical scavenging activity was higher in the coated apple slices compared to the control (Figure 1g). In our study, ABTS increased in the control samples, with a minimum of 1.92 ± 0.04 µmol TE g^−1^ FW at 4 days of cold storage and a maximum of 3.11 ± 0.05 µmol TE g^−1^ FW after 12 days (Figure 1f). The samples coated with CMC+SA showed the lowest variation from the control, although still statistically significant, with percentages of 51, 15, and 25% compared to the control in the three timings of storage, respectively (Figure 1f). A significant increment of ABTS was also recorded in the CMC+SA+CA and CMC+SA+OA samples, with values higher than the control of 57 and 74% after 4 days, 27 and 53% after 8 days, and 36 and 47% after 12 days of cold storage, respectively (Figure 1f). However, the ABTS of the CMC+SA+OA+CA samples was 2-fold higher (4.16 ± 0.11, 4.48 ± 0.02, and 5.53 ± 0.08 µmol TE g^−1^ FW) than that of the control samples throughout storage (Figure 1f).

In fresh-cut fruits, phenolic compounds and ascorbic acid have key roles in antioxidant activity [40]. The fresh-cut processing operation contributes to the increase in phenolic content and, consequently, antioxidant capacity of these products, as demonstrated in several fresh-cut fruits such as carrot, onion, potato, and dragon fruit [51,52,53,54,55,56,57,58,59,60,61]. The edible coatings retained the phenolic content during storage as well as improving antioxidant activity and slowing down qualitative decay [42]. Nicolau-Lapena et al. [45] demonstrated that ferulic acid with *Aloe vera* gel or alginate coating was effective in maintaining a high antioxidant content with an increase in the AOX by DPPH and FRAP values during storage in fresh-cut apple.

### 2.3. Antioxidant Enzymes

The wounding process also activate defense gene transcription in plant cells as well as an antioxidant defense system that includes several enzymes, such as superoxide dismutase, catalase, and all enzymes involved in the glutathione–ascorbate cycle, that scavenge reactive oxygen free radicals, involved in cell membrane damage due to lipid peroxidation [62]. As shown in Figure 2, the SOD and CAT activity increased significantly (*p* < 0.05) in all coated samples compared to the control sample. SOD activity showed the lowest values in the control samples (1.85 ± 0.05, 1.80 ± 0.07, and 2.28 ± 0.04 U g^−1^ FW) throughout storage (Figure 2a). SOD activity reached the maximum values of 3.33 ± 0.01 and 3.45 ± 0.02 U g^−1^ FW in the CMC+SA+OA- and CMC+SA+OA+CA-coated samples, respectively, at the end of the cold storage (Figure 2a). Lower values (2.37 ± 0.03 and 2.57 ± 0.03 U g^−1^ FW) were recorded in the CMC+SA- and CMC+SA+CA-coated samples after 12 days of cold storage, respectively (Figure 2a). 

Similarly, CAT activity was lower in the control samples, with values of 69.25 ± 0.94, 71.66 ± 1.24, and 74.83 ± 0.32 µmol g^−1^ FW as cold storage progressed (Figure 2b). CAT activity increased during cold storage in coated samples with the polysaccharidic bilayer (CMC+SA) and with the polysaccharidic bilayer combined by CA (TB), OA, and CA+OA, reaching the highest values of 94.58 ± 0.39 and 104.84 ± 2.35 µmol g^−1^ FW in the CMC+SA+OA- and CMC+SA+OA+CA-treated samples after 12 days of cold storage, respectively (Figure 2b). ROS produced in the fruit cell can be scavenged, or processed, by highly efficient antioxidant systems [63]. The ROS-processing systems, important for fruit metabolism, involve different enzymes, such as SOD and CAT, needed to modulate ROS levels. Cutting styles can modify cellular homeostasis, leading to the increase in ROS production in the site or adjacent to wounded cells. In fresh-cut pitaya fruits, it has been observed that an increase in antioxidant enzyme activity can finely modulate ROS during cold storage [61] and methyl salicylate-based pretreatment on fresh-cut pitaya can accelerate the ROS response against cutting process and regulate ROS levels by activating antioxidant enzymes, including SOD, APX, and CAT [64]. Kou et al. [65] demonstrated that ‘Huang guan’ pears treated with chitosan, pullulan, and calcium coating maintained higher SOD and CAT activity than uncoated samples, although a reduction in SOD and CAT activity decline in the peel during seven months of cold storage was observed.

**Figure 2 ijms-24-08315-f002:**
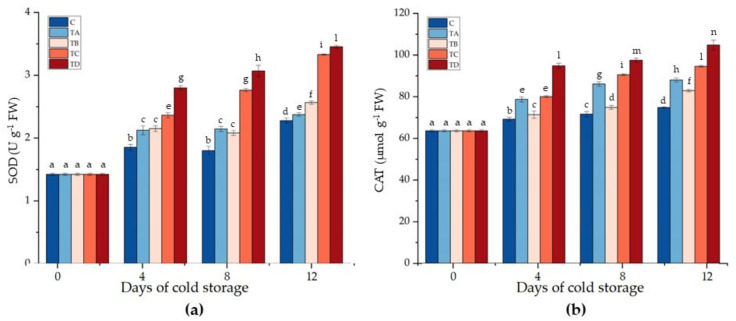
Superoxide dismutase (**a**) and catalase (**b**) activity of the control (C) and treated with CMC+SA (TA), CMC+SA+CA (TB), CMC+SA+AO (TC), and CMC+SA+CA+AO (TD) fresh-cut ‘Annurca Rossa del Sud’ apple after 0, 4, 8, and 12 days of cold storage at 4 °C ± 0.5. Error bars indicate standard deviation. Means followed by the same letter do not differ significantly at *p* = 0.05 (Duncan’s test).

The AA-GSH pathway takes part in the scavenging of ROS, whereas AA and GSH not only directly remove a variety of ROS but also carry out several other activities to keep the cytosol and other cellular organelles in a favorable state to increase the antioxidant capacity and lessen oxidative stress, which is brought on by various abiotic stresses [66]. 

The first enzyme of the AA-GSH cycle is APX, which converts hydrogen peroxide into water and monodehydroascorbate (MDHA), with AA acting as an electron donor. In our study, APX increased in all samples. The enzyme’s activity was lower in the control and CMC+SA+CA-coated samples, followed by the CMC+SA-coated samples, in all timings considered (Figure 3a). APX activity was higher in the CMC+SA+OA and CMC+SA+OA+CA samples, with values of 0.43 ± 0.02 and 0.53 ± 0.01 µmol g^−1^ FW after 4 days, 0.55 ± 0.01 and 0.63 ± 0.02 µmol g^−1^ FW after 8 days, and 0.63 ± 0.02 and 0.76 ± 0.02 µmol g^−1^ FW after 12 days of storage (Figure 3a). 

A part of MDHA is spontaneously converted into dehydroascorbate (DHA) and AA by monodehydroascorbate reductase (MDHAR). MDHAR activity followed an increasing trend in all samples. In particular, slight statistically significant variations were recorded in the CMC+SA- and CMC+SA+CA-coated samples, whereas in the CMC+SA+OA and CMC+SA+OA+CA apple slices, the enzyme activity was 1.6- and 2-fold higher than in the control at the end of the cold storage (Figure 3b). 

After this, DHA, by using GSH, produces GSSG through the action of dehydroascorbate reductase (DHAR). DHAR enzyme activity increased in all samples during cold storage. The control samples showed the lowest activity with values of 1.37 ± 0.01, 1.51 ± 0.01, and 1.66 ± 0.02 µmol kg^−1^ FW in the three timings of storage (Figure 3c). As with other enzymes, the highest activity was detected in the case of DHAR in the CMC+SA+OA+CA- coated samples, with values of 1.94 ± 0.03, 2.32 ± 0.04, and 2.63 ± 0.08 µmol kg^−1^ FW at 4, 8, and 12 days of cold storage (Figure 3c). 

Ultimately, employing NADPH as the electron donor, this GSSG regenerates GSH by the action of gluthatione reductase (GR). All treated samples showed an increase in GR activity compared to the control samples. In particular, the CMC+SA and CMC+SA+CA samples revealed an increase in GR activity around 11, 14, and 18% after 4, 8, and 12 days of cold storage, respectively (Figure 3d). In CMC+SA+OA and CMC+SA+OA+CA apple slices, after 12 days of cold storage, there was an increase in enzyme activity of 31 and 42%, respectively (Figure 3d). 

Similarly to our results, cellulose nanofibers increased the APX activity in fresh-cut ‘Red Fuji’ during 8 days of cold storage [67]. Carbossimethylcellulose with calcium chloride and liquid paraffin increased the activity of APX more than the control in pakchoi samples. The same increase was recorded for the other enzymes belonging to the AA-GSH cycle in treated pakchoi samples [68]. A coating composed of S-nitrosoglutathione-chitosan nanoparticles improved the MDHAR and DHAR activities in fresh-cut ‘Fuji’ apples throughout the whole storage time compared to control samples [69]. Similarly, Ackar et al. [70] demonstrated that a chitosan coating improved the AA-GSH cycle enzyme activities in fresh-cut Fuji apples more than the control samples.

**Figure 3 ijms-24-08315-f003:**
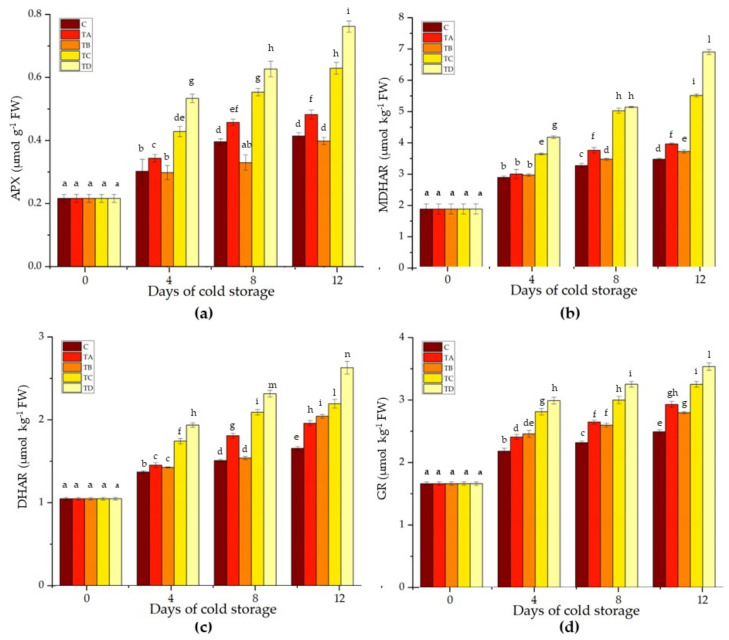
Ascorbate peroxidase (**a**), monodehydroascorbate reductase (**b**), dehydroascorbate reductase (**c**), and glutathione reductase (**d**) activity of the control (C) and treated with CMC+SA (TA), CMC+SA+CA (TB), CMC+SA+AO (TC), and CMC+SA+CA+AO (TD) fresh-cut ‘Annurca Rossa del Sud’ apple after 0, 4, 8, and 12 days of cold storage at 4 °C ± 0.5. Error bars indicate standard deviation. Means followed by the same letter do not differ significantly at *p* = 0.05 (Duncan’s test).

### 2.4. Enzymatic Browning and Oxidative Damage

Enzymatic browning is a natural phenomenon that occurs widely in minimally processed fruits and vegetables because of the bruising, cutting, and peeling stages. Enzymatic browning is caused by polyphenoloxidase (PPO), which is found in most fruits, vegetables, and certain seafood. Together with PPO, the related oxidative enzyme guaiacol peroxidase (GPX) may also cause enzymatic browning in fruits and vegetables [71]. In our study, GPX activity showed a statistically significant exponential increase in the control samples, with a maximum value of 121.30 ± 7.74 nmol g^−1^ FW after 12 days of cold storage (Figure 4). A slight significant increase was observed in all coated samples, although lower than the control samples. CMC+SA- and CMC+SA+CA-coated apples displayed the highest GPX values of 90.38 ± 1.06 and 101.88 ± 1.04 nmol g^−1^ FW at the end of cold storage, respectively. The CMC+SA+OA and CMC+SA+OA+CA apple slices revealed the efficacy of treatment with the lowest values of GPX activity (88.31 ± 0.28 and 79.80 ± 2.08 nmol g^−1^ FW) at the end of storage (Figure 4). In our study, PPO activity was probably delayed by coating application (Figure 5). The highest activity was recorded in the control samples (3.65 ± 0.14 µmol g^−1^ FW), while the lowest activity was observed in the CMC+SA+OA- and CMC+SA+OA+CA-coated samples (0.88 ± 0.02 and 1.01 ± 0.05 µmol g^−1^ FW) at the end of storage (Figure 5b). CMC+SA- and CMC+SA+CA-coated apple exhibited a PPO activity 3.3- and 2.5-fold lower (1.09 ± 0.07 and 1.41 ± 0.03 µmol g^−1^ FW) than the control samples at the last recorded timing (Figure 5b). In the presence of oxygen molecules, PPO catalyzes two distinct reactions: the hydroxylation of monophenols in *o*-diphenols and the oxidation of *o*-diphenols to *o*-quinones [71]. In this study, the *o*-diphenol content followed the same trend as PPO activity (Figure 5c). In particular, the content of *o*-diphenols was higher in the control samples, with an increasing percentage of 88, 114, 204, and 224% compared to CMC+SA-, CMC+SA+CA-, CMC+SA+OA-, and CMC+SA+OA+CA-treated samples at 12 days of cold storage, respectively (Figure 4 and Figure 6). The browning index (BI) was highest in the control samples, followed by CMC+SA and CMC+SA+CA, while the samples with a low browning index were CMC+SA+OA and CMC+SA+CA+OA (Appendix A; Figure 4).

Fresh-cut apple browning has been directly linked to PPO, although it also depends on the chemical structure and quantity of polyphenols [45]. In this study, coating application alleviated enzymatic browning by reducing PPO and GPX activities. According to our results, coatings composed of Acacia senegal, xanthan, and karaya gums inactivated PPO and GPX activity in fresh-cut apples [72]. Nicolau-Lapena et al. [45] reported that coatings with Aloe vera and alginate-calcium lactate delayed the enzymatic browning caused by PPO in apple discs. Fresh-cut apple cv Taaptimjaan coated with Aloe vera at 75% concentration displayed the lowest PPO and GPX activities after 6 days of cold storage [5]. Chitosan and chitosan–stevia coatings reduced the PPO activity in fresh-cut ‘Amasya’ apple, shielding the fruit from exposure to oxygen [73]. Similarly, chitosan nanoparticles reduced PPO activity but did not influence GPX activity in fresh-cut ‘Gala’ [74]. Cellulose nanofibers decreased the enzymatic browning caused by PPO and GPX activity in fresh-cut ‘Red Fuji’ during the cold storage [67].

LOX can have a positive or negative function in the response of the fruit to physical injury, depending on whether it is involved in autocatalytic peroxidation processes or as a signaling molecule that activates cellular defense [75,76]. Hydroperoxides produced by the action of LOX can cause tissue damage by inhibiting protein synthesis and altering cell membranes. Our results showed that the treatments significantly reduced the LOX activity (Figure 5d). Coating treatment influenced the LOX activity, which was 2-fold higher in the control samples compared to the CMC+SA+OA+CA apple slices at the end of cold storage (Figure 5d). The other treatments also improved the fruit’s resistance to lipoxidation; LOX activity in the control samples was 1.4-, 1.2-, and 1.7-fold higher than in the CMC+SA, CMC+SA+CA, CMC+SA+OA samples after 12 days of cold storage, respectively (Figure 5d). In line with our results, Wang et al. [67] reported that the coating formed by cellulose nanofibers improved the storability of ‘Red Fuji’ apples, inhibiting the LOX activity during cold storage. Moreover, arabic gum coating and γ-aminobutyric acid treatments (alone or in combination) significantly reduced LOX activity in Kinnow mandarins after 60 days of cold storage [19].

**Figure 4 ijms-24-08315-f004:**
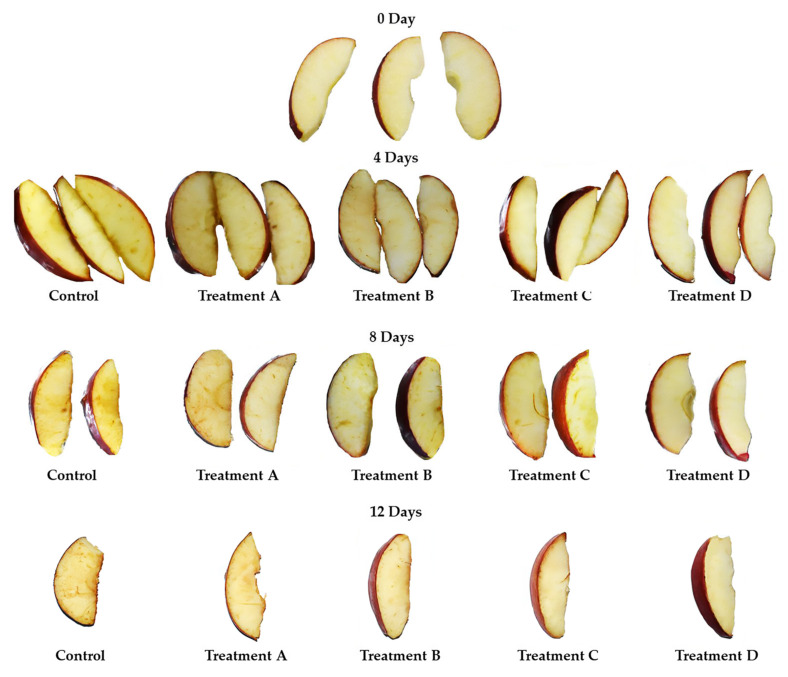
Browning surface in CMC+SA (TA), CMC+SA+CA (TB), CMC+SA+AO (TC), and CMC+SA+CA+AO (TD) in fresh-cut ‘Annurca Rossa del Sud’ apple after 0, 4, 8, and 12 days of cold storage at 4 °C ± 0.5.

**Figure 5 ijms-24-08315-f005:**
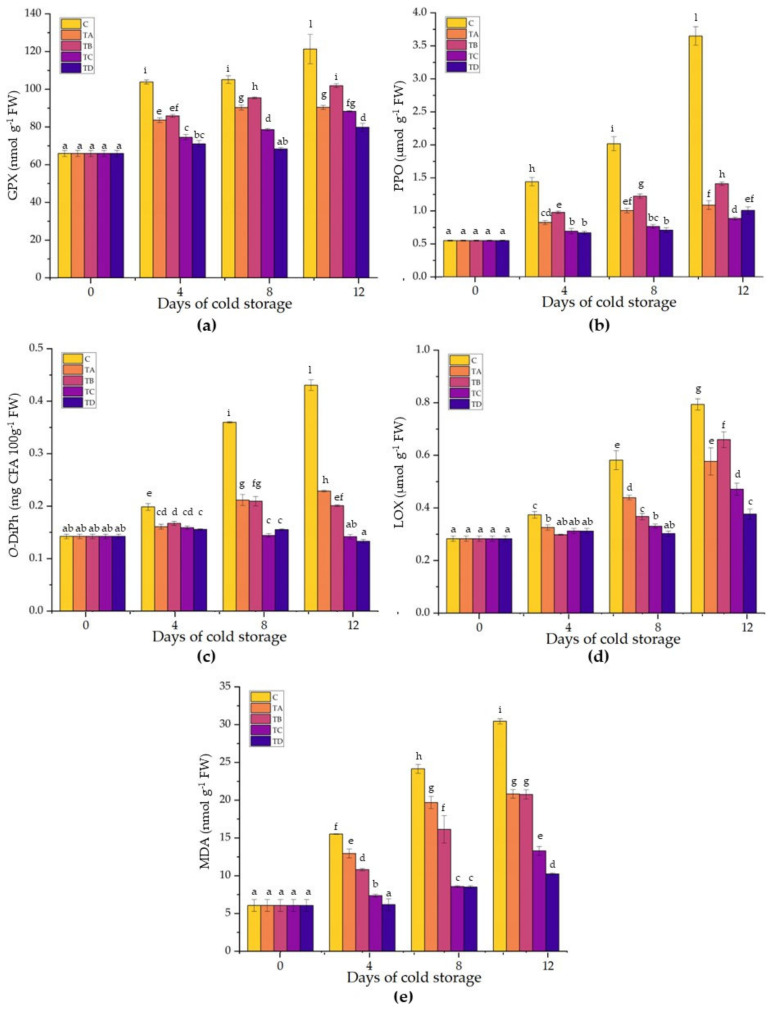
Guaiacol peroxidase (**a**), polyphenoloxidase (**b**), *ortho*-diphenols (**c**), lipoxygenase (**d**) and malondialdehyde (**e**) in control (C) and treated samples with CMC+SA (TA), CMC+SA+CA (TB), CMC+SA+AO (TC), and CMC+SA+CA+AO (TD) in fresh-cut ‘Annurca Rossa del Sud’ apple after 0, 4, 8, and 12 days of cold storage at 4 °C ± 0.5. Error bars indicate standard deviation. Means followed by the same letter do not differ significantly at *p* = 0.05 (Duncan’s test).

Cellular membrane lipid peroxidation is triggered by cold and stress, with MDA as a byproduct. MDA production is considered an ROS-induced biomarker during cold storage [77]. The MDA content was higher in the control samples than in the treated samples after 4, 8, and 12 days of cold storage. After 12 days of storage, the highest MDA content (30.45 ± 0.35 nmol g^−1^) was recorded in control samples, followed by CMC+SA and CMC+SA+CA-treated apples (20.83 ± 0.56 and 20.75 ± 0.62 nmol g^−1^), while the lowest MDA content was recorded in the CMC+SA+OA- and CMC+SA+OA+CA-treated samples (13.29 ± 0.58 and 10.24 ± 0.13 nmol g^−1^) (Figure 5e). The application of coating minimizes the generation of MDA by suppressing the accumulation of ROS by enhancing the accumulation of antioxidants in different fruit crops, such as apricot [78], persimmon [79], and sweet cherries [80]. Furthermore, cellulose nanofibril application reduced the MDA content in the treated apples by 1.34 times compared to control groups [67]. The chitosan coating, with or without heat treatment, retarded the increase in MDA content and maintained lower levels than control throughout storage time in ‘Gala’ treated samples [81]. Xin et al. [82] demonstrated that whey protein with TGase and sunflower oil film led to a reduction in MDA concentration by about 22% compared to uncoated apples. 

### 2.5. Statistical Analysis

A large multivariate dataset’s dimensionality might be reduced due to the principal component analysis (PCA). All the variance in the examined variables is explained by the identification of the principal components (PCs) (Figure 6). The first two principal components accounted for 88.48% of the variation (50.01 and 38.47% for PC1 and PC2, respectively) in treated and untreated minimally processed apple slices (Figure 6). CAT (r^2^ = 0.955), APX (r^2^ = 0.961), MDHAR (r^2^ = 0.959), DHAR (r^2^ = 0.981), GR (r^2^ = 0.964), POL (r^2^ = 0.764), DPPH (r^2^ = 0.816), ANT (r^2^ = 0.908), AA (r^2^ = 0.784), FLA (r^2^ = 0.845), FRAP (r^2^ = 0.954), and ABTS (r^2^ = 0.960) were positively correlated with PC1. In contrast, pH (r^2^ = 0.788), TSS (r^2^ = 0.972), WL (r^2^ = 0.943), MDA (r^2^ = 0.944), PPO (r^2^ = 0.925), LOX (r^2^ = 0.922), and *o*-DiPh (r^2^ = 0.925) were positively correlated with PC2, while TA (r^2^ = −0.855) negatively to one.

**Figure 6 ijms-24-08315-f006:**
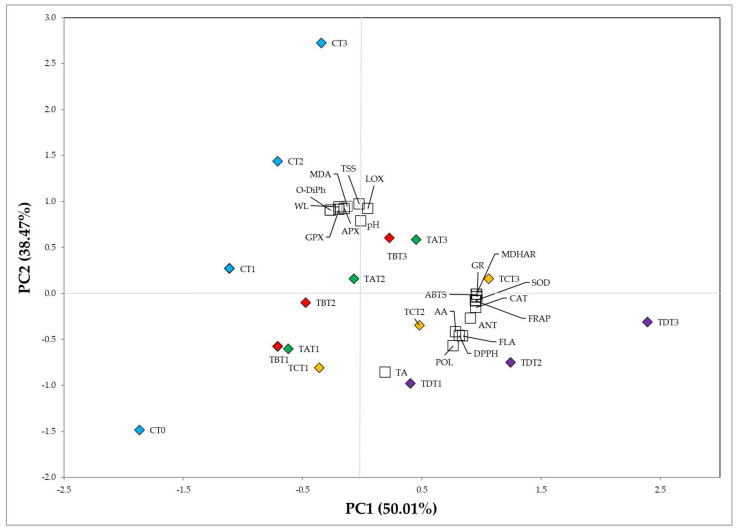
Principal component analysis on qualitative and biochemical traits in control (C, blue) and treated samples with CMC+SA (TA, green), CMC+SA+CA (TB, red), CMC+SA+AO (TC, yellow), and CMC+SA+CA+AO (TD, violet) in fresh-cut ‘Annurca Rossa del Sud’ apple after 0 (T0), 4 (T1), 8 (T2), and 12 days (T3) of cold storage at 4 °C ± 0.5.

The Pearson coefficient was used to conduct a correlation-based analysis, which revealed substantial negative (in red) and positive (in blue) connections between all examined attributes (Figure 7). The fruit’s TSS increases as it ripens and becomes sweeter and less acidic. The main elements used to assess the quality of food are pH, titratable acidity, and soluble solid levels [83]. In our study, pH showed a positive correlation with TSS (r = 0.768; *p* ≤ 0.01) and a negative connection with TA (r = −0.560; *p* ≤ 0.01). Pre-preparations, such as peeling, chopping, and slicing, cause serious mechanical damage to fresh-cut fruits and vegetables. These processing procedures improve the secondary metabolite production [40]. In this study, all the secondary metabolism components examined had positive correlations with one another: POL with FLA (r = 0.942; *p* ≤ 0.01), AA (r = 0.890; *p* ≤ 0.01), and ANT (r = 0.824; *p* ≤ 0.01) and subsequently with DPPH (r = 0.836; *p* ≤ 0.01), FRAP (r = 0.751; *p* ≤ 0.01), and ABTS (r = 0.720; *p* ≤ 0.01). Physiological conditions determine a low ROS generation in plant tissue. Wounding stress can compromise cellular homeostasis, increase respiration, and activate other ROS sources such as amine oxidases and NADPH oxidases, increasing ROS production around the injured cells [61]. In these cases, all antioxidant enzymes are activated. In our study, all antioxidant enzymes are positively correlated with each other: CAT with SOD (r = 0.923; *p* ≤ 0.01), APX (r = 0.945; *p* ≤ 0.01), MDHAR (r = 0.893; *p* ≤ 0.01), DHAR (r = 0.946; *p* ≤ 0.01), and GR (r = 0.943; *p* ≤ 0.01). The imbalance in ROS metabolism led to overactive lipoxygenase and the consequent accumulation of malondialdehyde, which contributes to oxidative damage to the cell membrane and affects storage quality. In addition, GPX- and PPO-induced enzymatic pathways can lead to the most damaging browning, which, in combination with lipid oxidation reactions, results in significant fruit loss [84]. In our study, we found a positive correlation between all the factors that determine fruit decay and browning. In fact, PPO showed a positive correlation between *o*-DiPh (r = 0.927; *p* ≤ 0.01) and MDA (r = 0.848; *p* ≤ 0.01) levels and GPX (r = 0.857; *p* ≤ 0.01) and LOX (r = 0.824; *p* ≤ 0.01) activities.

## 3. Materials and Methods

### 3.1. Fruit Sample and Experimental Design

‘Annurca Rossa del Sud’ apples were hand-harvested on 25th September in an experimental orchard located in Pignataro Maggiore (Caserta, Italy) at the CREA—Research Centre for Olive, Fruit and Citrus Crops (41°04′ N. 14°19′ E with an altitude of 61 m above sea) and were transported to the laboratory, screened for uniformity, appearance, and the absence of physical defects or decay. Fruits were placed in ‘melaio’ to redden for 20 days. The whole fruits were sanitized with sodium hypochlorite (1% *v/v*) and sterile water and sliced. Four solutions of carboxymethylcellulose (CMC) 1% *w/v*, sodium alginate (SA) 1% *w/v*, oxalic acid (OA) 0.5% *w/v*, and citric acid (CA) 1% *w/v* were prepared.

The apple slices were divided into five lots. The control lot was immersed in sterile water for 2 min, interspersed with 1 min of air drying. The second lot (TA) consisted of dipping, as for the control lot, in CMC and SA, the third lot (TB) in CMC, SA, and CA, the fourth lot (TC) in CMC, SA, and OA, and the fifth lot (TD) in all four solutions. The choice of all concentrations was determined by preliminary tests. At the end of coating application, the samples (n = 60, about 250 g each) were dried for 1 h at room temperature and stored in PET trays for ready-to-eat fruit at 4 ± 0.5 °C and 95% relative humidity. All analyses were carried out on three biological replicates at 0 (T0), 4 (T1), 8 (T2), and 12 (T3) days of cold storage, and three technical replicates were performed per biological replicate. Samples were frozen in liquid nitrogen and stored at −80 °C until further analysis.

### 3.2. Physicochemical Traits

For physicochemical analysis, the juice from the apple slices was obtained using a juice extractor (Hurom H-AA, Soul, Republic of Korea). The total soluble solid content (TSS) was assessed using a digital refractometer (Sinergica Soluzioni, DBR35, Pescara, Italy) and expressed as °Brix. Titratable acidity (TA) was carried out according to Magri et al. [85], by titrating 5 mL of diluted juice (1:5 *v/v*) with 0.1 mol L^−1^ NaOH and expressed as g of malic acid equivalent (MAE) per L of juice (g MAE L^−1^). The weight loss (WL) was established by calculating the difference between the initial weight from the final weight divided by initial weight at every sample timing, and the results were expressed as a percentage. The pH value was measured using a digital pH meter (Model 2001, Crison, Barcelona, Spain) at 25 °C. The slice colors were assessed using a Minolta colorimeter (CR5. Minolta Camera Co., Osaka, Japan) to determine chromaticity values L* (lightness), a* (green to red), and b* (blue to yellow). The browning index was calculated by converting the Hunter scale (Lab) values through the formula:(1)$$\mathrm{BI}=100-\sqrt{{\left(100-L\right)}^{2}}+{\;a}^{2}+{\;b}^{2}$$

The results were expressed as the percentage of browning (%).

### 3.3. Secondary Metabolites

Slices of apple were homogenized (1:10 *w/v*) in a hydroalcoholic solution (methanol/water 80:20 *v/v*). The hydroalcoholic extract was used to determine the secondary metabolite content. The total phenolic content (TP) was evaluated through the Folin–Ciocâlteu method described by Magri et al. [85], with some modifications. The mixture assay contained 20 µL hydroalcoholic extract, 100 µL Folin’s reagent, and sodium carbonate 7.5% *w/v*. The results were expressed as mg of gallic acid equivalent per 100 g^−1^ of fresh weight (mg GAE per 100 g^−1^ FW). The flavonoid content (TF) was assessed as reported by Morra et al. [86]. The results were expressed as mg of catechin equivalent (CE) per 100 g^−1^ FW. 

The ascorbic acid (AA) content was determined according to Malorni et al. [87], with some modifications. Apple fruits (1:5, *w/v*) were mixed with a solution of 16% *v/v* metaphosphoric acid and 0.18% *w/v* disodium ethylene diamine tetraacetic acid (Na-EDTA). The homogenate was centrifuged at 11,000× *g* for 10 min at 4 °C. The assay mixture was developed using 400 μL of extract, 0.3% *w/v* metaphosphoric acid, and diluted Folin’s reagent (1:5, *v/v*). The results were expressed as mg ascorbic acid (AA) per g^−1^ FW.

### 3.4. Antioxidant Activity

The antioxidant activity based on the DPPH method was measured according to Cirillo et al. [88], with slight modification. The reaction mix included DPPH 63.4 µM and 80 µL of hydroalcoholic extract. The AOX was expressed as µmol Trolox equivalent (TE) g^−1^ of FW. 

For the ABTS assay, the stock solutions consisted of 7.4 mM 2,2′-azino-bis (3-ethylbenzothiazoline-6-sulfonic acid) (ABTS^•+^) solution and 2.6 mM potassium persulfate solution. The working solution was then prepared by mixing the two stock solutions in equal quantities and allowing them to react for 12 h at room temperature in the dark. The solution was then diluted by mixing the ABTS^•+^ solution with phosphate-buffered saline 1X pH 8.0 to obtain an absorbance of about 0.8–0.9 units at 730 nm. The results were expressed as µmol TE g^−1^ of FW.

For the reducing power assay (FRAP), 100 µL of methanolic extract was mixed with a NaH_2_PO_4_/Na_2_HPO_4_ buffer 0.2 M, pH 6.0. Samples were incubated at 50 °C for 30 min. After this step, trichloroacetic acid solution 10% was added in the tubes, and they were placed on a shaker plate for 10 min. Subsequently, a sample of the reaction mixture was homogenated with ferric chloride 0.1%. The sample’s absorbance was monitored at 720 nm. The results were expressed as µmol TE g^−1^ of FW. 

### 3.5. Extraction for Enzymatic Activities

The TE crude extract was achieved by homogenizing tissue powder (1:3; *w/v*) with an extraction buffer containing 50 mM potassium phosphate buffer (pH 7.8), 1 mM sodium EDTA (pH 7), 6.25 mM polyethylene glycol (PEG), 5% (*w/w*) polyvinylpolypyrrolidone (PVPP), and 5 mM ascorbic acid, only for the APX enzyme extraction. The activities of catalase (CAT), superoxide dismutase (SOD), guaiacol peroxidase (GPX), ascorbate peroxidase (APX), monodehydroascorbate reductase (MDHAR), dehydroascorbate reductase (DHAR), and glutathione oxidoreductase (GR) were screened with this crude extract. 

The T_PPO_ crude extract was obtained by mixing apple powder (1:1.5 *w/v*) with 0.5 M sodium phosphate buffer pH 6.4 containing 2% (*w/v*) of PVPP. 

The T_LOX_ was realized by blending frozen apple tissue (1:3 *w/v*) with 50 mM potassium phosphate buffer pH 7.8, 1 mM sodium–EDTA pH 7.0, and 2% (*w/v*) of PVPP.

The protein content in all crude enzyme extracts examined was evaluated using the Bradford assay [89].

#### 3.5.1. Catalase and Superoxide Dismutase Activity

CAT (EC 1.11.1.6) activity was estimated as described by Magri and Petriccione, 2022 [90], with slight modifications. The reaction mixture contained 100 mM potassium phosphate buffer pH 7.0, 27 mM H_2_O_2_, and 100 μL of crude enzyme extract. The reaction was checked at 240 nm, and catalase activity was expressed as μmol of H_2_O_2_ per g FW. 

SOD (EC 1.15.1.1) was assessed as reported by Magri et al. [91], with some modifications. The reaction mixture contained 50 mM potassium phosphate buffer pH 7.8, 1.0 mM Na-EDTA pH 7.0, 13 mM methionine, 75 μM nitro blue tetrazolium chloride (NBT), 0.5 μM riboflavin, and 100 μL of proteic extract. The absorbance was registered at 560 nm and expressed as U g^−1^ FW.

#### 3.5.2. Ascorbate–Glutathione Pathway

All analysis was carried out as reported by Magri et al. [91], with slight modifications. APX (EC 1.11.1.11) activity was assayed using 100 mM potassium phosphate buffer pH 7.0, 0.33 mM ascorbic acid, 0.42 mM H_2_O_2_, 0.66 mM Na-EDTA pH 7.0, and 100 µL of crude enzyme extract. The reaction was monitored at 290 nm, and the enzyme activity was expressed as µmol of ascorbate g^−1^ of FW. 

MDHAR (EC 1.6.5.4) activity was executed using of 50 mM tris(hydroxymethyl)aminomethane hydrochloride pH of 7.6 (Tris-HCl), 2.5 mM sodium ascorbate, 0.1 mM reduced disodium salt hydrate (NADH), 0.25 U ascorbate oxidase, and 50 µL of crude enzyme extract. The absorbance was registered at 340 nm and expressed in mol kg^−1^ FW. 

DHAR (EC 1.8.5.1) activity was assessed by combining 50 mM potassium phosphate pH 7.0, 0.1 mM sodium EDTA pH 8.0, 2.5 mM reduced glutathione, 0.2 mM dehydroascorbate, and 50 µL of crude enzyme extract. The reaction was started by adding dehydroascorbate, and the increase in ascorbate was recorded at 265 nm. The results were expressed as mol kg^−1^ FW.

GR (EC 1.6.4.2) activity was carried out mixing 50 mM potassium phosphate pH 7.0, 0.5 mM glutathione disulfide (GSSG), 50 µL of crude enzyme extract, and 0.2 mM dihydro nicotinamide-adenine dinucleotide phosphate (NADPH). The absorbance was recorded at 340 nm and was expressed in mol kg^−1^ FW.

#### 3.5.3. Enzymatic Browning and Markers of Oxidative Damage

GPX (EC 1.11.1.7) activity was determined as described by Caracciolo et al. [92], with slight modification. The reaction mixture contained 100 mM potassium phosphate buffer pH 7.0, 1 mM sodium-EDTA pH 7.0, 8 mM H_2_O_2_, 15 mM guaiacol, and 100 μL of crude enzyme extract. GPX activity was monitored at 470 nm, and the results were expressed in nmol g^−1^ FW. 

PPO (EC.1.10.3.1) was defined as described by Battaglia et al. [93], with some modifications. The reaction mixture contained 0.5 M sodium phosphate buffer, pH 6.4, 0.07 M catechol, and 50 μL crude enzyme extract and the results were expressed as µmol g^−1^ FW. 

The *ortho*-diphenols content (*o*-DiPh) was carried out as reported by Vella et al. [94], with slight modifications. The hydroalcoholic extract was added to the reaction mixture consisting of 0.1 M HCl, 0.3 M sodium nitrite, 80 mM sodium molybdate dihydrate, and 0.2 M sodium hydroxide. The absorbance was recorded at 500 nm, and the results were expressed in mg caffeic acid 100 g^−1^ FW. 

LOX (EC. 1.13.11.34) was carried out as reported by Adiletta et al. [95], with slight modifications. The reaction was conducted in the presence of 0.1 M sodium phosphate buffer pH 6, 5 mM linoleic acid sodium salt, and 100 µL crude enzyme extract (10 µL). LOX activity was recorded at 234 nm and expressed as nmol hydroperoxides g^−1^ FW.

The MDA content was assessed as highlighted by Goffi et al. [96], with slight modifications. The extraction buffer contained 10% (*w/v*) trichloroacetic acid, 0.5% (*w*/*v*) thiobarbituric acid, and 0.25 N HCl. Frozen tissue (1:8 *w/v*) was homogenized with the extraction buffer, held at 95 °C for 30 min, and subsequently cooled on ice. The supernatant’s absorbance was measured at 450, 532, and 600 nm. The amount of MDA was expressed as nmol g^−1^ FW.

### 3.6. Statistical Analysis

Mean and standard deviation are the only data expressions (SD). Analysis of variance (ANOVA) and Duncan’s test at 5% significance level were used to analyze the mean differences between the minimally processed untreated apples and the coated apples. Different letters indicate differences considered significant at *p* < 0.05. Pearson’s correlation was used to examine the correlation coefficients between different characteristics (*p* < 0.05, *p* < 0.01). Principal component analysis (PCA) was used to explain how treatments affect physicochemical properties and antioxidant systems and to identify the principal components that account for most of the variance in the data set. Statistical analysis was performed using the SPSS software suite (version 20.0; SPSS Inc., Chicago, IL, USA) and OriginPro 2023 (OriginLab Corporation, Northampton, MA, USA).

## 4. Conclusions

In this study, a new treatment based on the combined effect of two beneficial and eco-sustainable compounds combined with two organic acids, such as malic and oxalic acid, have been evaluated. CMC and SA are alternately self-assembled on the surface of ‘Annurca Rossa del Sud’ apple slices and functionalized with organic acids. This study demonstrated that the combined coating was an efficient tool in controlling the qualitative, biochemical, and enzymatic parameters of fresh-cut apple fruit. The application of this coating led to a decrease in the weight loss, pH, and TSS of the samples, allowing the prolonged cold storage of apple slices. Furthermore, the coatings improved the non-enzymatic and enzymatic antioxidant defense system with a reduction in oxidative damages. The results demonstrated that CMC+SA+OA+CA significantly inhibited flesh browning, maintained lower color changes, PPO, LOX activity, and *o*-diphenol and MDA content of fresh-cut apple fruit compared to the control during 12 days of cold storage. The combined use of CMC+SA+OA+CA can be a valid and eco-friendly treatment for fresh-cut apple fruits to prolong their postharvest life during cold storage. Future experiments aim to test the microbiological properties of the layer-by-layer edible coating and consumer acceptability through the establishment of panel tests.

## Figures and Tables

**Figure 7 ijms-24-08315-f007:**
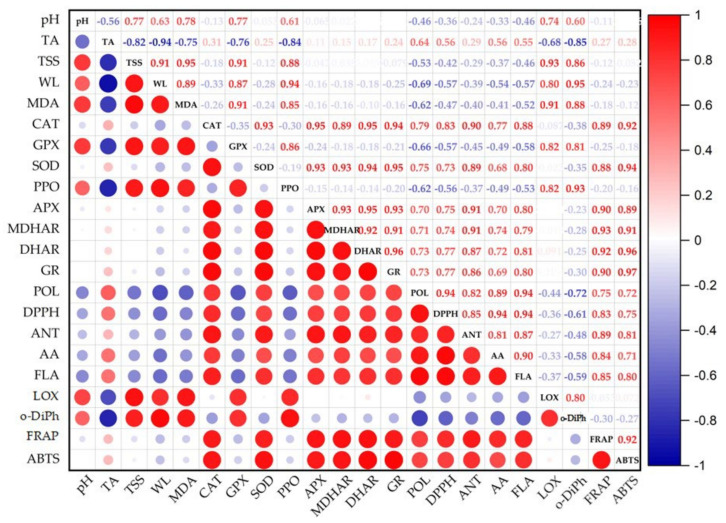
Pearson’s correlation matrix for qualitative and biochemical traits (*p* < 0.01).

**Table 1 ijms-24-08315-t001:** Physicochemical parameters (WL: weight loss; TSS: total soluble solid; TA: titratable acidity) of control (C) and treated with CMC+SA (TA), CMC+SA+CA (TB), CMC+SA+AO (TC), and CMC+SA+CA+AO (TD) fresh-cut ‘Annurca Rossa del Sud’ apple after 0 (T0), 4 (T1), 8 (T2), and 12 days (T3) of cold storage at 4 °C ± 0.5. The results are reported as mean ± standard deviation. Means followed by the same letter do not differ significantly at *p* = 0.05 (Duncan’s test).

Physicochemical Traits	Timing	C	TA	TB	TC	TD
WL (%)	T0	0 ± 0 a	0 ± 0 a	0 ± 0 a	0 ± 0 a	0 ± 0 a
	T1	2.36 ± 0.07 i	0.73 ± 0.03 de	0.57 ± 0.01 c	0.32 ± 0.01 b	0.20 ± 0.02 b
	T2	4.06 ± 0.10 l	1.39 ± 0.06 g	1.18 ± 0.03 f	0.72 ± 0.04 de	0.5 ± 0.02 c
	T3	4.94 ± 0.18 m	1.63 ± 0.03 h	1.41 ± 0.05 g	0.85 ± 0.01 e	0.72 ± 0.02 de
TSS (°Brix)	T0	11.3 ± 0.06 a	11.3 ± 0.07 a	11.3 ± 0.06 a	11.3 ± 0.06 a	11.3 ± 0.06 a
	T1	12.4 ± 0.03 g	11.6 ± 0.06 cd	11.6 ± 0.03 cd	11.4 ± 0.03 ab	11.4 ± 0.03 ab
	T2	12.8 ± 0.06 h	12.2 ± 0.10 f	12 ± 0.06 e	11.7 ± 0.06 d	11.5 ± 0.06 bc
	T3	13.6 ± 0.09 i	12.6 ± 0.10 h	12.4 ± 0.06 g	12.1 ± 0.06 f	11.9 ± 0.06 e
TA (g MAE L^−1^)	T0	9.28 ± 0.06 h	9.28 ± 0.06 h	9.28 ± 0.06 h	9.28 ± 0.06 h	9.28 ± 0.06 h
	T1	8.24 ± 0.08 c	9.28 ± 0.13 h	9.15 ± 0.06 efgh	9.31 ± 0.08 h	9.31 ± 0.08 h
	T2	7.87 ± 0.09 b	8.95 ± 0.07 def	9.01 ± 0.02 defg	9.25 ± 0.08 gh	9.18 ± 0.04 fgh
	T3	7.64 ± 0.16 a	8.81 ± 0.07 d	8.91 ± 0.04 de	9.11 ± 0.04 efgh	9.15 ± 0.06 efgh
pH	T0	4.06 ± 0.02 a	4.06 ± 0.02 a	4.06 ± 0.02 a	4.06 ± 0.02 a	4.06 ± 0.02 a
	T1	4.16 ± 0.02 de	4.15 ± 0.04 cde	4.16 ± 0.01 de	4.13 ± 0.01 bcd	4.06 ± 0.01 a
	T2	4.18 ± 0.01 de	4.17 ± 0.02 de	4.16 ± 0.01 de	4.16 ± 0.01 de	4.10 ± 0.01 ab
	T3	4.25 ± 0.01 g	4.23 ± 0.01 fg	4.19 ± 0.01 ef	4.19 ± 0.01 ef	4.11 ± 0.01 abc

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
