# Peer review of "Impact of Novel Active Layer-by-Layer Edible Coating on the Qualitative and Biochemical Traits of Minimally Processed ‘Annurca Rossa del Sud’ Apple Fruit"

_ijms, 2023, doi:10.3390/ijms24098315_

Round 1

Reviewer 1 Report

Dear author,

Thanks for your good work. My comments are in the attached file.

Regards

Author Response

Dear author,

Thanks for your good work. My comments are in the attached file.

Regards

Thank you for your suggestions to improve our manuscript. Please find attached the reply to your suggestions.

Best regards

Anna Magri

Reviewer 2 Report

Experimental work investigating the effect of edible coatings on the enzymatic darkening of sliced apples. The chapter contains all the necessary information based on a review of the literature on the subject. Specific comments below:

Methodology

In view of the fact that long storage of ripe fruit is associated with colour changes that may cause lack of consumer acceptance, did the authors consider adding such an analysis, e.g. according to the methodology of Cielab?

In the introduction to the article, the authors mention the safety (I assume microbiological) of sliced apples stored for long periods of time. Do the authors plan storage analyses in this area?

Conclusions

Please add a sentence or two on what analyses should be carried out to make the proposed coatings commercially applicable?

Author Response

Experimental work investigating the effect of edible coatings on the enzymatic darkening of sliced apples. The chapter contains all the necessary information based on a review of the literature on the subject. Specific comments below:

Methodology

In view of the fact that long storage of ripe fruit is associated with colour changes that may cause lack of consumer acceptance, did the authors consider adding such an analysis, e.g. according to the methodology of Cielab?

Cielab values and browning index data are included in Table S1 in the Supplementary Material.

In the introduction to the article, the authors mention the safety (I assume microbiological) of sliced apples stored for long periods of time. Do the authors plan storage analyses in this area?

Microbiological analyses are planned for the next experiments.

Conclusions

Please add a sentence or two on what analyses should be carried out to make the proposed coatings commercially applicable?

The requested information has been included in the revision manuscript.

Round 2

Reviewer 1 Report

Dear author,

Thanks for the revised file. My comments are in the attached manuscript.

Regards

Author Response

Dear Reviewer I have made the requested changes to the manuscript. 
